# COVID-19 Associated Guillain–Barré Syndrome: A Report of Nine New Cases and a Review of the Literature

**DOI:** 10.3390/medicina58080977

**Published:** 2022-07-22

**Authors:** Andreea Paula Ivan, Irina Odajiu, Bogdan Ovidiu Popescu, Eugenia Irene Davidescu

**Affiliations:** 1Department of Neurology, Colentina Clinical Hospital, 020125 Bucharest, Romania; andreea-paula.ivan@rez-umfcd.ro (A.P.I.); irina.odajiu@rez.umfcd.ro (I.O.); eugenia.davidescu@umfcd.ro (E.I.D.); 2Department of Clinical Neurosciences, “Carol Davila” University of Medicine and Pharmacy, 050474 Bucharest, Romania; 3Department of Cell Biology, Neurosciences and Experimental Myology, ”Victor Babeș” National Institute of Pathology, 050096 Bucharest, Romania

**Keywords:** Guillain–Barré syndrome, COVID-19, respiratory failure, polyneuropathies, peripheral neuropathies

## Abstract

Background: Guillain–Barré syndrome (GBS)—a rare condition characterized by acute-onset immune-mediated polyneuropathy—has been registered as a neurological manifestation of COVID-19, suggesting a possible link between these two conditions. Methods: We report a case series of patients with COVID-19-related GBS hospitalized in the Neurology Department of Colentina Clinical Hospital, Bucharest, Romania, between March 2020 and March 2021. Several variables were analyzed, such as the mean interval between the onset of COVID-19 symptoms and neurological ones, clinical features, treatment course, and outcome. Further on, we conducted a thorough literature review based on the PubMed and ScienceDirect scientific databases. Results: A total of 9 COVID-19 patients developed symptoms of GBS, out of which in 7, it manifested as an acute inflammatory demyelinating polyneuropathy (AIDP). Five patients presented respiratory failure, 2 requiring mechanical ventilation. All patients received a course of intravenous immunoglobulins, 2 additionally requiring plasma exchange. Upon discharge, all but 1 patient (who had not regained the ability to walk) had a positive outcome, and 1 died during admission. In the literature review, we analyzed the published sources at the time of writing. Conclusions: A link between COVID-19 and GBS might be possible; therefore, increased vigilance is required in the early identification of these cases for prompt diagnosis and treatment. Some notable differences such as an earlier onset of GBS symptoms, higher respiratory dysfunction, and higher mortality rates in COVID-19 patients have been observed between the presentation of GBS in the context of COVID-19 and GBS of other causes.

## 1. Introduction

Since its emergence in December 2019 in Wuhan, China, Coronavirus disease 2019 (COVID-19) has been spreading worldwide at an alarming pace, with 244,385,444 confirmed cases and 4,961,489 deaths at the time of writing, according to the World Health Organization [1]. As the name suggests, SARSV-CoV-2 (Severe Acute Respiratory Syndrome Coronavirus 2) predominantly affects the respiratory system, ranging from mild bilateral pneumonia to acute respiratory distress syndrome [2] due to an excessive increase in pro-inflammatory cytokine levels [3], after which in up to 97.7% of cases, pulmonary parenchyma alterations such as linear bands, ground-glass opacities, reticulations, bronchiolectasis, consolidations, bronchiectasis and volume loss may persist at least three months after the disease is detected by chest CT [4] or chest X-ray using a computational model [5]. However, numerous extra-respiratory manifestations (cardiac, gastrointestinal, hepatic, renal, neurological, olfactory, gustatory, ocular, cutaneous, and hematologic) have been reported as well [2,6]. In the acute phase, the most frequently occurring central nervous system manifestations of SARS-CoV-2 are headache and anosmia, whereas stroke and seizures are less commonly reported [7]. Regarding the peripheral nervous system, neuropathic pain accounted for between 7.24–42.8% of cases and is more frequently encountered [8], and Guillain–Barré syndrome (GBS) is much rarer [7]. 

Known as the most common cause of acute flaccid paralysis [8], GBS is a rare, heterogeneous condition characterized by acute-onset immune-mediated polyneuropathies. The underlying pathophysiologic mechanism is generally related to an abnormal immune response to previous infections (most typically, of the respiratory system), leading to autoantibody production responsible for peripheral nerve damage. These infectious agents can be either viral (e.g., human immunodeficiency virus, Cytomegalovirus, Epstein–Barr virus, Varicella-Zoster Virus, Herpes Simplex Virus, Hepatitis A, B, C, E Viruses, Chikungunya virus, Zika virus) or bacterial (e.g., *Campylobacter jejuni*, *Mycoplasma pneumoniae*, *Haemophilus influenzae*, *Escherichia coli*) [9]. 

GBS includes several variant subtypes from a clinical and electrophysiological standpoint. The most frequently occurring subvariants are acute inflammatory demyelinating polyradiculoneuropathy (AIDP) (85–90% cases), acute motor and sensory axonal neuropathy (AMSAN), acute motor axonal neuropathy (AMAN), and Miller–Fisher syndrome [10,11,12]. Several rarely occurring variants include Bickerstaff brainstem encephalitis, paraparetic GBS, acute pandysautonomia, purely sensitive GBS, facial diplegia, acute bulbar paralysis, and pharyngeal–cervical–brachial weakness [10,11,12]. 

Although a rarely occurring disease (1 to 2 cases per 100,000 per year) [13,14], some reports point towards what appears to be an increasing incidence in 2020, which coincides with the peak of the novel coronavirus outbreak [15,16]. In January 2020, Zhao et al. reported the first known case of GBS occurring in a 61-year-old woman infected with SARS-CoV-2 [17]. Furthermore, a study performed by Fragiel et al. on an Italian cohort over two months reported that GBS was more common among COVID-19 patients than among non-COVID 19 ones [18]. Since then, several similar cases have emerged, suggesting a possible link between the two conditions. Therefore, our report aims to outline whether the increased frequency of cases is a simple coincidence or there may be a possible link between GBS in COVID-19 patients.

## 2. Materials and Methods

We report a case series of nine new cases of COVID-19-related GBS among patients diagnosed and treated in the Neurology Department of Colentina Clinical Hospital, Bucharest, Romania, between March 2020 and March 2021. The patients’ characteristics, such as demographic ones, clinical manifestations, results of paraclinical investigations, and outcomes, were retrospectively extracted from the archived charts of the patients as well as the electronic internal hospital system, anonymized and stored in an Excell table. All categorical variables were stated as numbers and percentages. Mean and median values and the distribution of patient characteristics were calculated with SPSS Version 25.0 (SPSS Inc., Chicago, IL, USA).

In addition, we conducted a thorough literature review in May 2021 based on the PubMed and ScienceDirect scientific databases, using the terms “COVID-19 and GBS”, “Guillain–Barré syndrome following COVID-19 infection”, and “GBS as a manifestation of COVID-19 infection” as search elements. All full-length articles written in English, including patients diagnosed with GBS and COVID-19 available at the moment of writing, were included, and an analysis of the clinical and paraclinical characteristics was made. As a result, 30 articles were included: 1 multi-center observational study, 6 case series, 1 scientific letter, 3 letters to the editor, and 19 case reports (Figure 1). We acknowledge that many other cases of GBS and COVID-19 patients could have been omitted due to their inclusion in more general studies on neurological manifestations in COVID-19 patients, which we did not encompass in our report. 

## 3. Results

Nine patients (seven males and two females) with a median age of 56 years (varying from 39 to 67) were included in this case series, with an incidence of 1.42% (9 out of 631 hospitalized patients with the co-occurrence of COVID-19 and neurological manifestations). All patients had at least one positive nasopharyngeal swab RT-PCR test for SARS-CoV-2 during hospitalization. Unfortunately, SARS-CoV-2 sequencing to determine which SARS-CoV-2 variant each patient had was not possible due to the laboratory possibilities. However, we presume that most of our patients had the Alpha variant according to the reports of the National Institute of Public Health from Romania in that period [19]. The most common COVID-19 symptoms were fever, cough, and myalgia. The interval between the onset of COVID-19 symptoms and that of neurological ones ranged from 1 to 21 days (median interval = 9.5 days), suggesting a predominantly para-infectious mechanism of appearance of GBS. The period between the onset of neurological symptoms and hospitalization ranged from 24 h to 14 days. Typical GBS symptoms upon admission consisted in ascending lower limb weakness (7 cases), general areflexia (5 cases), and paresthesia (7 cases). 

### 3.1. Presentation

#### 3.1.1. Clinical Presentation

The characteristics of the clinical presentation of GBS are presented in Table 1. The median mMRC (Modified Medical Research Council) score was 2 for the lower limbs and 3.5 for the upper limbs, typical for the ascending pattern of GBS. Two patients developed paraplegia during hospitalization. Five patients presented cranial nerve involvement such as facial palsy, nystagmus, dysarthria, and mixed dysphagia. While hyporeflexia and areflexia were predominant, there was one case of hyperreflexia (without any history of upper motor neuron lesion). Regarding dysautonomia, one patient suffered from paralytic ileus, while two presented urinary and fecal incontinence. In terms of GBS variant distribution, there was a notable predominance of the demyelinating form (AIDP) among our patients (7 cases—77.7%), with only two cases of AMAN (22.3%). Signs of respiratory failure were present among five of our patients, with two of them requiring mechanical ventilation. 

#### 3.1.2. Paraclinical Investigations

Further explorations such as blood tests, cerebrospinal fluid (CSF) analysis, and antiganglioside antibody tests were performed to confirm the diagnosis of GBS Lumbar puncture in seven out of the nine cases. CSF fluid evaluation revealed cyto-albuminologic dissociation in all but one of the tested cases. Serum ganglioside antibodies were evaluated for a single patient and were negative. Blood tests were performed on admission, during hospitalization, and shortly before discharge. Their mean values are listed in Table 2. Inflammatory (e.g., C reactive protein (CRP), interleukin-6 (IL-6), ferritin, fibrinogen), and coagulation (international normalized ratio (INR), D-dimers) markers were elevated among most patients. Thoracic CT scans were performed to evaluate the severity of the COVID-19 infection. The median pulmonary parenchymal surface affected by reticulonodular infiltration (indicative of SARS-CoV-2-related pneumonia) was 20%. Due to exceptional conditions, electrophysiological investigations could not be performed, as our department was the first one declared and dedicated only to COVID-19 patients, and specific paraclinical investigations could not be accessed.

#### 3.1.3. Treatment

All patients received a course of intravenous immunoglobulins (0.4 g/kg per day for five days), two requiring plasma exchange (before the course of intravenous immunoglobulins). The SARS-CoV-2 infection was managed with antivirals such as remdesivir or favipiravir (five patients) and immunosuppressants such as tocilizumab (one patient) [20] and hydroxychloroquine (one patient) [21] according to local guidelines, as directed by the infectionist. Additional therapy with dexamethasone was administered to patients with clinically significant pulmonary involvement (four cases). All patients received low molecular weight heparin (LMWH), depending on patient particularities (eight received prophylactic doses, and one received therapeutic doses).

#### 3.1.4. Outcome

Eight patients were discharged within 2 to 52 days (median interval = 12 days), with a single case of exitus due to acute respiratory failure by day two of hospitalization. All subjects showed motor and sensory function improvement in variable degrees. Upon discharge, only one patient had not regained the ability to walk. Cranial nerve involvement also improved, except for a single case of persisting horizontal nystagmus. By the time of discharge, all patients tested negative for SARS-CoV-2. The first positive and the first negative RT-PCR interval ranged from 1 to 22 days (median interval = 14 days). None of the patients developed signs of COVID-19-related long-term complications such as chronic respiratory dysfunction, neutropenia, or thrombocytopenia (Table 3).

### 3.2. Literature Review

Regarding the existing literature that reported the association between COVID-19 and GBS: 19 case reports, 6 case series, 3 scientific letters, 1 scientific letter, and 1 observational multi-center study—all comprising a total of 91 patients were included at the moment of the article writing. Out of the 91 reported patients, a male predominance (70.32%) was encountered as in GBS of other etiologies [11]. Their ages ranged from 17 to 84 years (median age = 57 years). Most reported studies used RT-PCR as the primary diagnostic tool for detecting SARS-CoV-2 infection (91.9%). One case, reported by Esteban Molina et al., had a negative RT-PCR test, which was later considered a false negative result, while Svačina et al. reported an equivocal RT-PCR result [22,23]. Three patients were diagnosed with COVID-19 using thoracic computer tomography and one via serologic tests. 

The most notable COVID-19 symptoms were fever, cough, dyspnea, and myalgia. The interval between the onset of COVID-19 symptoms and that of neurological ones varied from 12 h to 36 days (median = 11 days). On the other hand, the most frequently occurring neurological symptoms were progressive tetraparesis (55 patients—60.4%), areflexia (61 patients—67.03%), paresthesia (39 patients—42.85%), hypoesthesia (32 patients—35.16%), and facial weakness (17 patients—18.68%). Respiratory failure was noted in 24 cases (26.37%), 18 requiring mechanical ventilation, the description being provided in Table 4.

Several investigations have been performed, including the cerebrospinal fluid examination (CSF), electromyographic studies, MRI, and serum antiganglioside antibodies assessment. Out of the 91 cases, 49 underwent CSF evaluation during hospitalization. Thirty-two patients presented cyto-albuminologic dissociation, a finding that may suggest a more rapidly installing pattern of CSF fluid abnormalities in the COVID-19 related GBS, as opposed to non-COVID-19 induced GBS, in which only up to 50% of cases or less have modified CSF results within the first two weeks of disease and 75% after the first three weeks [10]. Nerve conduction studies were performed on 81 patients. Their descriptions are listed in Table 5. 

MRI was reported for 24 patients, which revealed leptomeningeal enhancement in one case, facial nerve enhancement in another, and caudal nerve roots enhancement in two subjects. Serum antiganglioside antibodies were evaluated in 15 subjects, out of which four were positive. GBS variants were reported in 77 out of the 91 cases, and their distribution follows—56 AIDP cases (72.52%), seven AMAN cases (7.7%), seven AMSAN cases (7.7%), one case of polyneuritis cranialis (1.1%), four cases of Miller–Fisher syndrome (4.4%), and two cases of axonal-demyelinating sensorimotor polyradiculoneuropathy (2.2%). 

The vast majority of cases were treated with a course of intravenous immunoglobulins (74 patients—81.31%). Ten underwent plasma exchange (10.98%), one received both therapeutic options, one patient received only corticosteroids, and three received no treatment. In terms of outcome, there were five cases of exitus (5.49%), while the remaining 74 reported cases displayed variable degrees of improvement. 

## 4. Discussion

According to our case series that included nine patients with GBS and COVID-19, the occurrence of GBS might be correlated to COVID-19. This could be potentially associated with the high incidence of respiratory failure, as in our patients (five out of nine and two requiring mechanical ventilation), which was higher than that reported in the literature (26.37%), suggesting a possible correlation between GBS secondary to COVID-19 and an increased risk of respiratory dysfunction versus GBS of other causes (25%) [13,15,28]. In addition, it would have been interesting to determine if the degree of respiratory failure differs depending on the SARS-CoV-2 variant since it is generally acknowledged that the Omicron variant induces less severe pulmonary involvement and consequently less respiratory failure, for which additional studies are needed [51]. In such a way, it would be easier to understand if the respiratory failure is related only to GBS or if the SARS-CoV-2 infection also has an additional contribution to it. 

In addition, the fact that the interval between COVID-19 onset and GBS symptoms onset in our case series as well as in other reports was less than two weeks [27,32,35,37,39,40] might indicate that GBS following COVID-19 occurs slightly earlier than GBS following other infections, in which symptoms arise two to four weeks after infection [13,23]. These figures also suggest a predominantly para-infectious mechanism of appearance, considering the interval between the onset of COVID-19 symptoms and neurological ones (1 to 21 days)—slightly shorter than literature reports [18,20]. Potential mechanisms involved in this para-infectious pattern, also noted by several other authors [27,52], can be attributed either to a highly pro-inflammatory state (frequently occurring in COVID-19) leading to a dysimmune reaction, as seen in Zika Virus–GBS-related cases, or to direct viral nerve destruction [53,54]. Moreover, the significant response to intravenous immunoglobulins might again indicate an underlying immune-mediated mechanism of COVID-19-related GBS rather than direct peripheral nerve damage. On the other hand, the mortality rate may be higher among COVID-19-related GBS patients than in non-COVID-19 GBS cases, less than 3% [48,55].

Another curiosity observed in our case series was the presence of hyperreflexia in one patient, which is rarely encountered in GBS [13]. 

Regarding similitudes between GBS in COVID-19 patients and GBS of other causes, the demyelinating pattern, in other words, AIDP on electrophysiologic studies, was predominant in our case series as well as in other reports [47,50], similarly to non-COVID–19-related GBS [10], analogous to the male predominance [11]. 

### Limitations

Due to circuit restrictions related to the SARS-CoV-2 pandemic, our clinic has drastically limited access to diagnostic tests for SARS-CoV-2 infected patients, such as electrophysiologic studies, causing difficulties in distinguishing GBS from COVID 19-related respiratory complications and other conditions such as critical illness neuropathy. Moreover, the prevalence of GBS related to COVID-19 is yet to be determined, and given the impaired access to medical services associated with the COVID-19 pandemic, mild cases of GBS are at risk of being overlooked. Thus, further studies should be performed to establish the prevalence and understand the underlying mechanisms of this co-occurrence. Additionally, physicians should maintain a high level of suspicion when dealing with GBS cases and screen them for SARS-CoV-2 infection, given that COVID-19-related respiratory symptoms may be mild or even absent in many instances.

The increased incidence of Guillain–Barré Syndrome cases and the causal association with COVID-19 is not enough for this relationship to be established with certainty, as long as other known and described associations were not excluded, as investigating for other diseases or co-infections like syphilis, hepatitis, tuberculosis, Lyme disease, influenza, systemic lupus erythematosus, etc., was not possible due to restrictions and special conditions, even though clinical manifestations were clear and the differential diagnosis did not need to be made exhaustively.

## 5. Conclusions

Considering recent worldwide case reports, there might be a link between GBS and COVID-19. Our report reiterated the numerous similarities between GBS related to COVID-10 and GBS of other causes, such as the predominance of demyelinating forms, male predominance, and the favorable response to intravenous immunoglobulins. The differences reside in an earlier onset of GBS symptoms, higher respiratory dysfunction, and higher mortality rates in COVID-19 patients compared to GBS due to other causes, making prompt diagnosis and treatment crucial in managing such cases.

## Figures and Tables

**Figure 1 medicina-58-00977-f001:**
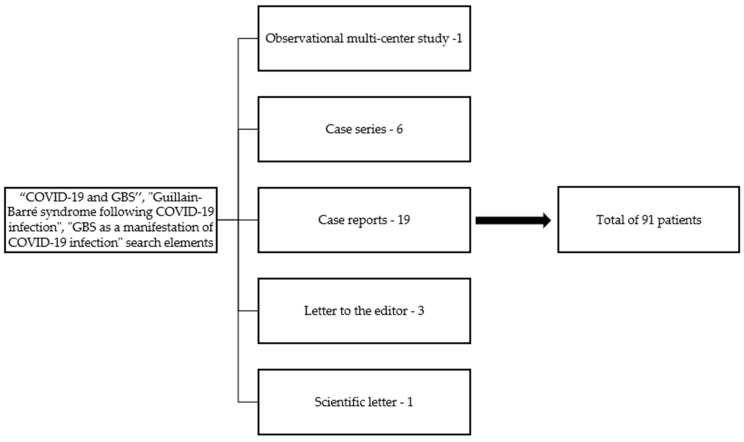
Article selection.

**Table 1 medicina-58-00977-t001:** Patient clinical characteristics.

Case	Sex	Age	Comorbidities	COVID-19 Symptoms	SaO2 (%) on Admission	Days between the Onset of Neurological Symptoms and Admission to Hospital	Days between COVID-19 Symptoms and Neurological Symptoms	Neurological Symptoms
1	M	56	-	Dyspnea, cough myalgia	96% with 3 L O2/min	8	11	Progressive ascending tetraparesis (1/5 LL, 3/5 UL), hyperreflexia, bilateral facial palsy, paresthesia, superficial and deep hypoesthesia, dysautonomia (paralytic ileus, urinary, and fecal incontinence)
2	F	65	Arterial hypertension, type 2 diabetes mellitus, obesity	Dyspnea	95% with 2 L O2/min	3	-	Progressive ascending tetraparesis followed by paraplegia (0/5 LL, 1/5 UL), areflexia, paresthesia, lower limbs deep anesthesia, dysarthria, mixed dysphagia
3	M	67	-	Dyspnea, fever, chills	99% on room air (o.r.a)	3	1	Progressive asymmetric paraparesis (LL 2.5/5), left upper limb weakness, lower back pain, areflexia, paresthesia, lower limbs superficial and deep hypoesthesia, dysphonia, constipation
4	M	56	Arterial hypertension	Fever, cough	97% o.r.a	14	14	Progressive ascending paraparesis up to knee level (4/5), areflexia in lower limbs, “glove and stocking” paresthesia, lower limbs hypoesthesia
5	F	56	-	Cough, myalgia	96% o.r.a	7	7	Progressive ascending flaccid tetraparesis (1/5 LL, 3/5 UL), areflexia, paresthesia, superficial hypoesthesia, deep anesthesia in lower limbs, dysphonia, mixed dysphagia, fecal and urinary incontinence
6	M	41	Obesity, Hypercholesterolemia history of AMSAN	-	98% o.r.a	1	-	Progressive ascending flaccid tetraparesis (3/5 LL, 4/5 UL), hyporeflexia
7	M	51	Arterial hypertension, obesity	Fever, cough, myalgia, chills, diarrhea	97% o.r.a	7	21	Progressive ascending tetraparesis (3/5 LL, 4/5 UL), areflexia
8	M	39	Stroke, central core disease, type 2 diabetes mellitus	Headache, loss of appetite	98% o.r.a	8	6	Progressive ascending tetraparesis (3/5 LL, 4/5 UL), ataxia, lower limbs areflexia, upper limbs paresthesia, dysarthria
9	M	51	Epilepsy secondary to childhood meningoencephalitis	Hyposmia, hypogeusia, dyspnea	97% o.r.a	3	7	Progressive ascending tetraparesis followed by paraplegia, areflexia, paresthesia, superficial and deep hypoesthesia, left side peripheral facial palsy, exhaustible right-beating nystagmus, mixed dysphagia, dysarthria

Abbreviations: o.r.a—on room air; UL—upper limbs; LL—lower limbs.

**Table 2 medicina-58-00977-t002:** Mean blood test results.

	Admission	Mid-Hospitalization	Discharge
White blood cell count (×1000/µL)	9.2644	10.7500	7.3460
Lymphocyte count (×1000/µL)	1.5344	2.2700	3.0940
Platelet count (×100/µL)	342.5556	290.4286	308.0000
D-dimers (µg/mL)	0.6475	1.3967	1.0040
Fibrinogen (mg/dL)	566.2500	514.3333	518.7500
INR	1.0613	1.1467	1.0400
C-reactive protein (mg/L)	29.1722	89.7267	14.8040
Ferritin (ng/mL)	571.8333	642.7500	1410.6667
Procalcitonin (ng/mL)	0.0600	0.2200	0.1167
IL-6 (pg/mL)	26.5133	24.0367	19.6400
LDH (IU/L)	233.1667	347.0000	222.3333

**Table 3 medicina-58-00977-t003:** Diagnosis, treatment, outcome.

	GBS Variant	CSF Findings	Ganglioside Autoantibodies	Thoracic CT—% of Pulmonary Involvement	Acute Respiratory Failure	Need for Intubation	G.B.S. Treatment	COVID-19 Treatment	Hospitalization (Days)	Outcome
1	AIDP	Normal	Absent	70%	Yes	No	IVIG	Remdesivir, LMWH 8000 IU, DEXA	32	Significant motor function improvement—mMRC 2/5 LL, 4/5 UL, improved paresthesia
2	AIDP	CAD	-	20%	Yes	Yes	PEX, IVIG	Remdesivir, LMWH 400 IU, DEXA	36	Gradual motor function improvement—mMRC 1/5 LL, 3/5 UL
3	AIDP	CAD	Absent	50%	Yes	No	IVIG	Remdesivir, LMWH 4000 IU, DEXA	18	Gradual motor function improvement (can ambulate with unilateral support)
4	AIDP	CAD	-	15%	No	No	IVIG	Favipiravir, LMWH 4000 IU	10	absent motor weakness, improved paresthesia, and hypoesthesia
5	AIDP	-	-	20%	Yes	No	IVIG	LMWH 6000 IU	2	Exitus due to respiratory failure
6	AMAN	-	-	15%	No	No	IVIG	LMWH 4000 IU	5	Favorable
7	AMAN	CAD	-	50%	No	No	IVIG	Tocilizumab LMWH 4000 IU	12	Favorable
8	AIDP	CAD	-	15%	No	No	IVIG	Hydroxycloroquine LMWH 4000 IU	9	residual tetra-ataxia (predominantly in the lower limbs), can ambulate with unilateral support, no sensitive impairment
9	AIDP	CAD	-	45%	Yes	Yes	PEX, IVIG	LMWH 4000 IU, DEXA	52	improved motor function (cannot ambulate), residual hypoesthesia, residual nystagmus, postural and intentional tremor, dysphonia

Abbreviations: AIDP—Acute inflammatory demyelinating polyneuropathy; AMAN—Acute motor axonal neuropathy; CAD—cytoalbuminologic dissociation; DEXA—dexamethasone; IVIG—intravenous immunoglobulins; mPRED—methylprednisolone; PEX—plasma exchange; LMWH—low molecular weight heparine.

**Table 4 medicina-58-00977-t004:** Clinical features synopsis in literature.

Article	Type of Study	No. of Cases	Mean Age	Sex	COVID-19 Symptoms	Days between COVID-19 Symptoms and Neurological Symptoms	Neurological Symptoms
Bueso et al. [24]	Case report	1	60	F	Fever, cough, myalgia, dysgeusia	22	Symmetrical tetraparesis, inability to walk, lower limbs areflexia, UL hyporeflexia, sacro-lumbar pain, paresthesia dysautonomia (fluctuations in HR and mean arterial pressure, fecal incontinence, urinary retention)
Kamel et al. [25]	Case report	1	72	M	Fever, cough, myalgia, dyspnea	21	Tetraparesis, unsteady, stamping gait, global areflexia, “glove and stocking” hypoesthesia, sensory ataxia, hypoesthesia to fine touch, and vibration distal to clavicle
Singh et al. [26]	Case report	1	45	M	Fever, cough, dyspnea	7	Tetraparesis, LL areflexia, UL hyporeflexia, bilateral facial paresis, paresthesia, hypoesthesia to fine touch, and vibration distal to calf
Sidig et al. [27]	Case report	1	65	M	Fever, cough, sore throat, headache, generalized fatigue	5	Ascending tetramelic tetraparesis and paresthesia, generalized hypotonia and areflexia, truncal weakness, bilateral papilledema, olfactory nerve involvement, bilateral facial nerve involvement, slight palatal muscle weakness, facial paresthesia with the inability to close his mouth and his both eyes, urinary incontinence
Webb et al. [28]	Case report	1	57	M	Fever, cough, myalgia, headache, malaise, diarrhea	7	Progressive tetraparesis, hypotonia, areflexia, foot dysesthesia, and hypoesthesia
Korem et al. [29]	Case report	1	58	F	Fever, cough, back pain	14	Tetraparesis, unstable gait, persistent back pain radiating to both lower extremities, foot paresthesia, hyporeflexia
Almutairi et al. [30]	Case report	1	36	M	Cough, diarrhea	14	Bilateral progressive distal extremity numbness, LL areflexia, UL hyporeflexia, tetramelic hypoesthesia, speech impairments, bilateral facial weakness
Mantefardo et al. [31]	Case report	1	17	F	Dyspnea, right flank pain, vomiting	N/A	Paraplegia, areflexia
Camdessanche et al. [32]	Case report	1	64	M	Fever, dyspnea	11	Flaccid tetraparesis, paresthesia in feet and hands, global areflexia, dysphagia
Sedaghat et al. [33]	Case report	1	65	M	Cough, fever, occasional dyspnea	14	Ascending symmetric tetraparesis, global areflexia, bilateral facial paresis, reduction in the vibration, and fine touch sensation distal to the ankle joints
Virani et al. [34]	Case report	1	54	M	Cough	10	Ascending tetraparesis, global areflexia, LL paresthesia
Zhao et al. [17]	Letter to editor	1	61	F	Fever, cough	8 days after neurologic symptoms onset	Progressive tetraparesis, LL areflexia, distal hypoesthesia
Caamaño et al. [35]	Case report	1	61	M	Fever, cough	10	Peripheral bilateral facial paresis
Coen et al. [36]	Letter to editor	1	70	M	Cough, myalgia, generalized fatigue	10	Paraparesis, distal allodynia, urinary retention, constipation
Alberti et al. [37]	Case report	1	71	M	Fever	7	Progressive flaccid tetraparesis, paresthesia, global areflexia
Yakoby et al. [38]	Case report	1	35	M	Fever, cough	9	Tetramelic motor deficit, left LL areflexia, right LL hyporeflexia, intermittent fasciculations, UL coarse resting tremor, LL sensory deficit to light touch, and pinprick
Diez-Porras et al. [39]	Case report	1	54	M	Fever, cough, myalgia	5	Severe flaccid tetraparesis, global areflexia, bilateral facial palsy, dysphagia, hypoesthesia in the left mandibular region, and distal UL region
Rajdev et al. [40]	Case report	1	36	M	Fever, respiratory distress, cough, chills, myalgia	18	Ascending tetraparesis, difficulty walking, LL areflexia and paresthesia, UL hyporeflexia,
Rana et al. [41]	Case report	1	54	M	Fever, rhinorrhea, odynophagia, chills, night sweats	14	Ascending tetraparesis, areflexia, facial diplegia, ophthalmoparesis, resting tachycardia, urinary retention
Mackenzie et al. [42]	Case report	1	39	F	Ageusia, anosmia, headache, myalgia	20	Ascending tetraparesis, inability to walk, generalized areflexia, left arm paresthesia
Molina et al. [22]	Scientific letter	1	55	F	Fever, dry cough, dyspnea	14	Progressive tetraparesis, areflexia, hands and feet paresthesia, intense lumbar pain irradiating to both legs, dysphagia, bilateral facial diplegia, eyelid closing weakness, tongue, and perioral paresthesia
Yiu et al. [43]	Case report	1	69	M	Hypoxemic respiratory failure, fever, headache, dyspnea, decreased oral intake, diarrhea	36	Symmetric flaccid tetraparesis, generalized areflexia
Chan et al. [44]	Case series	2	1. 68 2. 84	2M	1.- Fever, upper respiratory symptoms 2.- Fever	1–18 2–23	Patient 1: Progressive gait disturbance and hands and feet paresthesia, weakness, inability to ambulate, bilateral facial weakness, dysphagia, dysarthria, neck flexion Patient 2: Progressive arm weakness, progressive gait disturbance (inability to stand or ambulate independently), UL hyporeflexia, LL areflexia, bilateral facial paresis, hands, and feet paresthesia, toes diminished vibration and proprioception, autonomic dysfunction
Okhovat et al. [45]	Case series	6	56	4F 2M	Fever (4 patients), dyspnoea (4 patients), cough (2 patients), malaise (1 patient), headache (1 patient)	14	Ascending tetraparesis (6 patients), areflexia (5 patients), facial weakness (1 patient), paresthesia (3 patients), hypoesthesia (5 patients)
Gutierrez-Ortiz et al. [46]	Case series	2	1. 50 2. 39	2M	1. fever, cough, malaise, headache, low back pain, anosmia, ageusia 2. fever, diarrhea, generally poor condition	Patient 1 and 2: 3	Patient 1: Gait instability, vertical diplopia, perioral paresthesia Patient 2: Diplopia
Foresti et al. [47]	Case series	17	53	11M 6F	N/A	12 h–28 days	N/A
Toscano et al. [48]	Case series	5	61	4M 1F	Fever (3), cough (4), hypogeusia (2), anosmia (2), sore throat (1), asthenia (1)	5–10	Tetraplegia (2 patients), tetraparesis (2 patients), areflexia (5 patients), bulbar symptoms (1 patient), facial weakness (1 patient), facial diplegia (1 patient), paresthesia (3 patients)
Nanda et al. [49]	Case series	4	55	3M 1F	Fever (2), abdominal pain (1), cough (2), sore throat (1)	8.5	Progressive tetraparesis (3 patients), paraparesis (1 patient), generalized areflexia (2 patient), bilateral facial palsy (1 patient), hypoesthesia (1 patient)
Svačina et al. [23]	Letter to editor	3	68	2M 1F	Patient 1: Reduction of general condition, dyspnea Patient 2: Rhinorrhea, headache Patient 3: dyspnea	Patient 1 and 2: 4 Patient 3: N/A	Patient 1: Progressive flaccid tetraparesis, general areflexia, phrenic-bulbar involvement Patient 2: Facial palsy, oculomotor palsy, hypoglossal palsy Patient 3: Tetraparesis, areflexia, distal paresthesia
Abu-Rumeileh et al. [50]	Observational multicenter study	30	59 0.2	22M 8F	Fever, cough, dyspnea, dysgeusia, anosmia, gastrointestinal symptoms	Median 16–35	Tetraparesis—25, predominant paraparesis—1, predominant upper limb paresis—3 + sensory symptoms in all patients

Abbreviations: HRheart rate; LL—lower limbs; UL—upper limbs.

**Table 5 medicina-58-00977-t005:** Diagnosis, treatment, and outcome synopsis in literature.

Article	GBS Variant	CSF Findings	Electromyographic Findings	Spinal MRI.	Serum Antiganglioside Antibodies	Respiratory Dysfunction/ Need for Intubation (+/−)	Treatment for G.B.S.	Outcome
Bueso et al. [24]	N/A	CAD	-	N/A	N/A	Yes/(−)	IVIG	Respiratory improvement, ambulating with assistance, persistent LL neuropathic pain
Kamel et al. [25]	AIDP	CAD	Decreased velocity, decreased CMAP, less prominent focal slowing, significantly delayed late responses	Degenerative changes	Negative anti-GD1a and anti-GQ1b antibodies	No	IVIG	After one month: no motor weakness (patient could walk without support), marked improvement in deep sensation examination
Singh et al. [26]	AMSAN	CAD	Decreased amplitude, normal distal latencies, and prolonged F-wave latencies, decreased recruitment, decreased response in sensory and motor nerves	Degenerative changes	N/A	No	IVIG	Motor and sensory function improved significantly
Sidig et al. [27]	AIDP	N/A	Predominant demyelination pattern	Normal	N/A	Yes/(+)	IVIG	Exitus after seven days due to progressive respiratory failure
Webb et al. [23]	AIDP	CAD	Decreased velocity, prolonged distal motor latencies in motor and sensory nerves, dispersed motor action potentials, prolonged or absent F-waves, reduced velocities in sensory nerves, absent sensory action potentials in the right median nerve	N/A	N/A	Yes/(+)	IVIG	N/A
Korem et al. [29]	N/A	CAD	-	Degenerative changes	-	No	IVIG	About 80% improvement in motor strength of all limbs, complete resolution of paresthesia
Almuitairi et al. [30]	N/A	Normal	Absent SNAP of median and superficial peroneal nerves, delayed distal motor latency, borderline low CMAP of right median nerve, slow conduction velocities, prolonged F-wave latencies	N/A	N/A	No	IVIG	After one month: mild residual bilateral lower motor neuron facial weakness
Mantefardo et al. [31]	N/A	CAD	-	-	-	Yes/N/A	No	Exitus due to respiratory failure
Camdessanche et al. [37]	AIDP	CAD	Demyelinating pattern	-	Negative	Yes/(+)	IVIG	N/A
Sedaghat et al. [33]	AMSAN	-	Decreased amplitude CMAP, no response for SNAP, decreased recruitment	-	-	No	IVIG	N/A
Virani et al. [34]	N/A	-	-	Normal	-	Yes/(+)	IVIG	Residual LL weakness
Zhao et al. [17]	AIDP	-	Delayed distal latencies, absent F waves	-	-	No	No	Symptoms completely resolved
Caamaño et al. [27]	N/A	CAD	-	-	-	No	Low dose oral prednisone for 2 weeks	After two weeks: barely notable improvement on both sides
Coen et al. [36]	AIDP	CAD	Sensorimotor demyelinating polyneuropathy with “sural sparing pattern”, decreased persistence, or absent F-waves	Normal	Negative	No	IVIG	Rapid improvement
Alberti et al. [28]	AIDP	CAD	Absent sural nerve SNAP, absent tibial nerve CMAP, markedly increased common peroneal CMAP distal latency, markedly decreased velocity, moderately decreased CMAP amplitude (with spatial and temporal dispersion) for the same nerve, decreased ulnar SAP amplitude	-	-	Yes/(−)	IVIG	Patient died due to respiratory failure
Yakoby et al. [38]	N/A	CAD	-	Normal	Negative	No	IVIG	Noticeable improvement
Diez-Porras et al. [39]	AIDP	CAD	Conduction blocks, absence of F waves in the right ulnar nerve and axon potentials in the F response of the right tibial nerve) of diffuse distribution	-	IgM for GM2 and GD3	Yes/(+)	IVIG	Seven weeks later: the patient was able to walk independently with support
Rajdev et al. [40]	AIDP	CAD	Patchy, multifocal demyelination with unequivocal conduction block, prolonged F-wave latencies, sural sparing	Normal	N/A	Yes/(+)	IVIG	Before discharge: motor strength continued to improve (muscle strength of 4/5 in proximal and 5/5 in distal muscle groups bilaterally) following extubation
Rana et al. [41]	AIDP, Miller Fisher syndrome	-	Consistent with the demyelinating form of GBS with secondary axonal degeneration	Normal	-	Yes/(+)	IVIG	N/A
Mackenzie et al. [42]	N/A	CAD	Prolonged distal motor latencies and action potential amplitudes, sural sparing	Degenerative changes	-	Yes/N/A	P.E.X.	Discharged after 20 days with the improvement of neurological status (LL—4/5(MRC)) and respiratory symptoms
Molina et al. [21]	AIDP	CAD	Prolonged distal motor latencies, absent F waves in the posterior tibial or cubital nerves, prolonged distal latencies in the left and right facial nerves with potential time dispersion. Potential desynchronization of the sensory nerve trunks of the arms with reduced velocities	Slightly enhanced leptomeningeal uptake in brainstem and cervical spinal cord level	-	No	IVIG	Motor balance 5/5 of the right arm, 3/5 of the left arm and 4/5 of both legs, persistence of paresthesia
Yiu et al. [43]	N/A	CAD	Prolonged sensory and motor amplitudes, widespread, reduced sensory nerve and CMAP amplitudes, slowing of conduction velocities in the left median and ulnar nerves with severe, demyelinating range slowing of left tibial NCV, reduced recruitment	-	-	Yes/(+)	IVIG	Fully independent patient 100 days after GBS diagnosis and able to walk 3.21 Kilometers independently with frequent breaks
Chan et al. [44]	Patient 1—N/A	CAD	-	Normal	Negative	No	PEX.	After 28 days: dysphagia has resolved, can ambulate with minimal assistance
	Patient 2—N/A	CAD	-	N/A	Elevated GM2 IgG/IgM antibodies	Yes/(+)	PEX IVIG	N/A
Okhovat et al. [45]	AIDP (2) AMAN (2) AMSAN (2)	CAD (1)	Consistent with GBS (6)	Normal (3)	Negative (1)	Yes/(+) (1) 2. No (5)	IVIG (3 patients) PEX (3)	Favorable (6 patients)
Gutierrez-Ortiz et al. [46]	MFS.	Patient 1: CAD	-	-	Positive for anti- GD1b IgG antibody	No	IVIG	Marked improvement
Foresti et al. [47]	AIDP (16)	N/A	Consistent with GBS (16)	N/A	N/A	N/A	IVIG (15) PEX (2)	Discharged (16 patients) Death (1 patient)
Toscano et al. [48]	AMSAN (2) AMAN (1) AIDP (2)	Normal (2)	Consistent with GBS (5)	Enhancement of caudal nerve roots (2), Enhancement of the the facial nerve (1), normal (2)	Negative (3 patients)	Yes (3)/(2+) No (2)	IVIG (5) PEX (1)	After four weeks: Two patients remained in the ICU. Two patients had flaccid paraplegia One patient was walking independently
Nanda et al. [49]	AMAN (2) AIDP (1) AMSAN (1)	CAD	Consistent with GBS (4)	Degenerative changes (4)	N/A	Yes (1)/(+) No (3)	IVIG (4)	1 patient: death 3 patients: significant improvement
Svačina et al. [22]	Patients 1, 3: Axonal-demyelinating sensorimotor polyradiculoneuropathy Patient 2: polyneuritis cranialis	CAD (3)	Consistent with GBS (3)	N/A	Patient 1: anti-sulfatide IgM autoantibodies Patient 2, 3: negative	Yes (1) N/A(2)	IVIG (3)	Patient 1: Sever tetraparesis Patient 2: Mild oculomotor palsy Patient 3: Severe tetraparesis
Abu-Rumeileh et al. [50]	Classical GBS (27), Facial diplegia (1), Pure sensory form (1), Pharyngeal-cervical-brachial (1)	CAD—7 Normal—14	AIDP (23), AMAN(2) Equivocal (5)	N/A	N/A	Yes (5)	PEX(2), IVIG(25), None(3)	Response to treatment(23)

Abbreviations: AIDP—Acute inflammatory demyelinating polyneuropathy; AMAN—Acute motor axonal neuropathy; AMSAN—acute motor and sensory axonal neuropathy; CMAP—Compound Muscle Action Potential; MFS—Miller Fisher syndrome CAD—cytoalbuminologic dissociation; IVIG—intravenous immunoglobulins; NCV—nerve conduction velocity; PEX—plasma exchange; SNAP—Sensory Nerve Action Potentials.

## Data Availability

The datasets generated during the analysis of this study are available from the corresponding author on reasonable request.

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
