# Peer review of "COVID-19 Associated Guillain–Barré Syndrome: A Report of Nine New Cases and a Review of the Literature"

_medicina, 2022, doi:10.3390/medicina58080977_

Round 1

Reviewer 1 Report

1) Abstract. L 27-29. Conclusions: A link between COVID-19 and GBS might be possible; therefore, increased vigilance  is required in the early identification of these cases for prompt diagnosis and treatment. Some notable differences have been observed between the presentation of GBS in the context of COVID-19  and GBS of other causes. Please summarise few notable differences between the presentation of GBS in the context of COVID-19  and GBS of other causes.

2) Introduction.  L 38-40. Although SARSV-CoV-2 (Severe Acute Respiratory Syndrome Coronavirus 2) predominantly affects the respiratory system, numerous extra-respiratory manifestations (cardiac, gastrointestinal, hepatic, renal, neurological, olfactory, gustatory, ocular, cutaneous, and hematologic) have been reported as well [2]. Please Improve the description of predominantly consequences on respiratory system and add these references:

A- A Novel Computational Model for Detecting the Severity of Inflammation in Confirmed COVID-19 Patients Using Chest X-ray Images. Diagnostics 202111, 855. https://doi.org/10.3390/diagnostics11050855

B- Interstitial Lung Disease at High Resolution CT after SARS-CoV-2-Related Acute Respiratory Distress Syndrome According to Pulmonary Segmental Anatomy. J. Clin. Med. 202110, 3985. https://doi.org/10.3390/jcm10173985

C- Cytokine Profiles as Potential Prognostic and Therapeutic Markers in SARS-CoV-2-Induced ARDS. J. Clin. Med. 202211, 2951. https://doi.org/10.3390/jcm11112951

D-  Characteristics, Management, and Outcomes of Elderly Patients with Diabetes in a Covid-19 Unit: Lessons Learned from a Pilot StudyMedicina 202157, 341. https://doi.org/10.3390/medicina57040341

3) Introduction. L61-67. Although a rarely occurring disease (1 to 2 cases per 100,000 per year) [10, 11], some  reports point towards what appears to be an increasing incidence in 2020, which coincides with the peak of the novel coronavirus outbreak [12, 13]. In January 2020, Zhao et al. reported the first known case of GBS occurring in a 61-year-old woman infected with SARS-  CoV-2 [14]. Furthermore, a study performed by Fragiel et al. on an Italian cohort over two months reported that GBS was more common among COVID-19 patients than non-COVID  19 ones [15]. Since then, several similar cases have emerged, suggesting a possible link  between the two conditions. Please underline the novelty of the study.

4) 3.1. Presentation 95 3.1.1. Clinical presentation. Please summarise and report here the most important clinical characteristics of patients.

5) 5. Conclusions L236-242. Considering recent case reports worldwide, there might be a link between GBS and COVID-19. Numerous similarities between this co-occurrence and GBS of other causes have been noted, such as the predominance of demyelinating forms and the favorable response to intravenous immunoglobulins. Nevertheless, some notable differences from the typical GBS occurrences have been pointed out, including higher respiratory dysfunction  and higher mortality rates, making prompt diagnosis and treatment crucial in managing  such cases. Please underline the novelty of the study and the clinical implication of observations.

Author Response

We do thank the reviewer for the precious time alocated to our work and for precious recommendations.

Reviewer 1

1) Abstract. L 27-29. Conclusions: A link between COVID-19 and GBS might be possible; therefore, increased vigilance is required in the early identification of these cases for prompt diagnosis and treatment. Some notable differences have been observed between the presentation of GBS in the context of COVID-19 and GBS of other causes. Please summarise few notable differences between the presentation of GBS in the context of COVID-19 and GBS of other causes.

This issue was addressed in lines 29-31.

2) Introduction.  L 38-40. Although SARSV-CoV-2 (Severe Acute Respiratory Syndrome Coronavirus 2) predominantly affects the respiratory system, numerous extra-respiratory manifestations (cardiac, gastrointestinal, hepatic, renal, neurological, olfactory, gustatory, ocular, cutaneous, and hematologic) have been reported as well [2]. Please Improve the description of predominantly consequences on respiratory system and add these references:

A- A Novel Computational Model for Detecting the Severity of Inflammation in Confirmed COVID-19 Patients Using Chest X-ray Images. Diagnostics 2021, 11, 855. https://doi.org/10.3390/diagnostics11050855

B- Interstitial Lung Disease at High Resolution CT after SARS-CoV-2-Related Acute Respiratory Distress Syndrome According to Pulmonary Segmental Anatomy. J. Clin. Med. 2021, 10, 3985. https://doi.org/10.3390/jcm10173985

C- Cytokine Profiles as Potential Prognostic and Therapeutic Markers in SARS-CoV-2-Induced ARDS. J. Clin. Med. 2022, 11, 2951. https://doi.org/10.3390/jcm11112951

D-  Characteristics, Management, and Outcomes of Elderly Patients with Diabetes in a Covid-19 Unit: Lessons Learned from a Pilot StudyMedicina 202157, 341. https://doi.org/10.3390/medicina57040341

Thank you for the suggestion, we have addressed the issue in lines 40-46 and added the suggested references.

3) Introduction. L61-67. Although a rarely occurring disease (1 to 2 cases per 100,000 per year) [10, 11], some reports point towards what appears to be an increasing incidence in 2020, which coincides with the peak of the novel coronavirus outbreak [12, 13]. In January 2020, Zhao et al. reported the first known case of GBS occurring in a 61-year-old woman infected with SARS-  CoV-2 [14]. Furthermore, a study performed by Fragiel et al. on an Italian cohort over two months reported that GBS was more common among COVID-19 patients than non-COVID  19 ones [15]. Since then, several similar cases have emerged, suggesting a possible link between the two conditions. Please underline the novelty of the study.

We have addressed the issue in lines 76-78.

4) 3.1. Presentation 95 3.1.1. Clinical presentation. Please summarise and report here the most important clinical characteristics of patients.

The most important clinical characteristics of the patients are already summarized in paragraph 3.1.1 (lines 115-125) as well as detailed in Table 1.

5) 5. Conclusions L236-242. Considering recent case reports worldwide, there might be a link between GBS and COVID-19. Our report reiterated the fact that numerous similarities have been noted between GBS related to COVID-10 and GBS of other causes, such as the predominance of demyelinating forms and the favorable response to intravenous immunoglobulins. Whereas the differences reside in an earlier onset of GBS symptoms, higher respiratory dysfunction and higher mortality rates in COVID-19 patients in comparison to GBS due to other causes, making prompt diagnosis and treatment crucial in managing such cases. Please underline the novelty of the study and the clinical implication of observations.

Thank you for the recommendation the issue was addressed in lines 262-265.

Reviewer 2 Report

Introduction:

-          The section about neurological complications of COVID-19 could be expanded. Cohort studies reported neurological complications such as stroke, myelitis, and peripheral neuropathies in a significant percentage of patients during the acute phase of the disease. Several studies also reported neuropathic pain during the acute phase of the disease (Jena et al., 2022; Oguz-Akarsu et al., 2022).

Methods:

-          Research search: please specify any filters and limits used.

-          Data extraction: describe any assumptions made about any missing or unclear information.

-          The inclusion of a flow diagram, showing numbers of studies screened, assessed for eligibility, and included in the final analysis may improve the quality of the manuscript.

Results

-          Results of the literature review should be reported in this section

Author Response

We do thank the reviewer for the time allocated to our work and for very useful advice in improving our manuscript.

Reviewer 2

Introduction:

-          The section about neurological complications of COVID-19 could be expanded. Cohort studies reported neurological complications such as stroke, myelitis, and peripheral neuropathies in a significant percentage of patients during the acute phase of the disease. Several studies also reported neuropathic pain during the acute phase of the disease (Jena et al., 2022; Oguz-Akarsu et al., 2022).

      Thank you for the suggestion, we have addressed the issue in lines 50-52.

Methods:

-          Research search: please specify any filters and limits used.

        We have addressed this issue in lines 93-97.

-          Data extraction: describe any assumptions made about any missing or unclear information.

         We have addressed this issue in lines 97-100.

-          The inclusion of a flow diagram, showing numbers of studies screened, assessed for eligibility, and included in the final analysis may improve the quality of the manuscript.

        We kindly thank for the suggestion a flow diagram was added – Flow diagram 1 on page 12.

Results

-          Results of the literature review should be reported in this section

        Thank you very much for the recommendation, we have addressed the issue in lines 164-202.

We would also like to mention that all changes were done using track changes and that we revised once again the references and included one study in the analysis of literature, which was by mistake omitted in the first writing and modified accordingly the results of the literature analysis.